# Influence of DMSO Non-Toxic Solvent on the Mechanical and Chemical Properties of a PVDF Thin Film

António Diogo André [1,2], Ana Margarida Teixeira [1,2] and Pedro Martins [1,3,*]

1 Associated Laboratory of Energy, Transports and Aeronautics (LAETA), Biomechanic and Health Unity (UBS), Institute of Science and Innovation in Mechanical and Industrial Engineering (INEGI), 4200-465 Porto, Portugal; a.diogo.andre@icloud.com (A.D.A.)
2 Faculty of Engineering, University of Porto (FEUP), 4200-465 Porto, Portugal
3 Aragon Institute for Engineering Research (i3A), Universidad de Zaragoza, 50018 Zaragoza, Spain
* Correspondence: palsm@fe.up.pt

**Abstract:** Piezoelectric materials such as PVDF and its copolymers have been widely studied in different areas and with promising applications, such as haptic feedback actuators or deformation sensors for aided-mobility scenarios. To develop PVDF-based solutions, different protocols are reported in the literature; however, a toxic and harmful solvent is commonly used (dymethilformamide (DMF)). In the present study, a non-toxic solvent (dymethilsulfoxide (DMSO)) is used to dissolve PVDF powder, while a specific ionic liquid (IL), [PMIM][TFSI], is used to enhance piezoelectric properties. A PVDF/IL thin film is characterized. The physical material characterization is based on optical analysis (to ensure the sample's homogeneity) and on mechanical linear behaviour (Young's modulus of 144 MPa and yield stress of 9 MPa). Meanwhile, a chemical analysis focuses on the phase modifications introduced by the addition of IL ($\beta$ phase increase to 80% and a degree of crystallinity, $\chi$, of 30%). All the results obtained are in good agreement with the literature, which indicates that the proposed experimental protocol is suitable for producing PVDF-based thin films for biomedical applications.

**Keywords:** PVDF thin film; polymeric material; optical analysis; mechanical characterization; chemical modifications

## 1. Introduction

Smart materials are currently a subject of high research value. They have been studied across different research fields, and there are significant applications in areas such as aerospace and aeronautics [1,2], medicine and rehabilitation [3–5], bioengineering [6,7], electronics [8] and architecture and design [9]. Despite their current applications, some of these materials were developed to mimic the human muscular and nervous systems [10] using non-biological materials. Given their ability to respond to external factors, these materials can be classified according to the input stimuli [11–13]. Known inputs can be thermal [14,15], magnetic [14,16], light [17,18], pH [19,20], pressure [10,19], moisture [21] or electrical [14,22]. Materials reacting to an electrical input, in particular the piezoelectric subclass, are well-known and are frequently used for healthcare purposes and biomedical applications [23], such as in nanomedicine and in tissue engineering scaffolds. Some examples of common materials used are zinc oxide (ZnO), cellulose nano fibril, boron nanotubes and lead zirconate titanate (PZT) [24].

Polyvinylidene fluoride (PVDF) and its copolymers have excellent properties that make them the ideal choice in several applications from electronics to bioengineering and medical applications [25,26]. PVDF is a crystalline polymer with good chemical and weather resistance as well as good distortion and creep resistance at low and high temperatures [27]. Moreover, it is flexible, light (low density), biomimetic, biocompatible and easily processable. Such characteristics make it a good solution for muscle-like actuators [28]. Other

relevant properties of PVDF are a large dielectric constant of 10 ($\epsilon'$) [29], high polarity, ionic conductivity of around $10^{-4}$ S/cm at 20 °C [30], high piezoelectric coefficient of 49.6 pm/V [31] and high mechanical strength (Young's modulus $\approx$ 1.6 GPa and yield strength $\approx$ 45 MPa [32]). Chemically, there is evidence of at least five polymorphic modifications in its structure ($\alpha$, $\beta$, $\gamma$, $\delta$ and $\epsilon$), with each phase being responsible for different behaviours in the material, caused by intrinsic properties. For example, $\beta$ and $\gamma$ phases are the most desirable when the goal is to use PVDF as a sensor or soft actuator due to their piezoelectric, pyroelectric and ferroelectric abilities [33,34]. These phases, mainly the $\beta$ phase, can be promoted using different methodologies, including mechanical stretching of $\alpha$ phase [35], melting PVDF under specific and controlled conditions (i.e., high pressure) [36], applying an external electrical field [37], using ultra-fast cooling [38], solution crystallization at temperatures below 70 °C [39] or by addition of nucleating fillers (e.g., ionic liquids (ILs)). The inclusion of ILs (typically composed of an organic cation and an organic/inorganic anion) brings other advantages, such as lower actuation voltage and improvement to electromechanical stability and durability, leading to higher material performance [32]. Moreover, some solvents, such as dimethyl formamide (DMF), dimethylacetamide (DMAC) and dimethylsulfoxide (DMSO) can also play a role in $\beta$ phase formation [40], since higher dipole moments tend to favour the formation of phases with piezoelectric properties [41].

Several authors have studied this polymeric material for many applications with different purposes [42]. For example, Mat Nawi et al. [43] used PVDF with polyethylene glycol dissolved in DMAC to improve the process of water treatment. In the field of medicine, Wang et al. [44] used PVDF films as the sensory component in a sensor system to monitor breathing and heartbeats during sleep. Moreover, PVDF has already been investigated as an energy harvester by Hu et al. [45]. They proposed a design that uses a PVDF film to capture energy from bending as a power source for pacemakers.

Focusing on PVDF's properties, Correia et al. [32,46] have studied the mechanical, chemical and electrical properties of PVDF-based materials with consideration of the inclusion of different ILs and using DMF as a solvent. Despite using different process techniques, both studies focus on the influence of different ILs on the properties of the material. All ILs increased the electroactive (EA) $\beta$ phase content in proportion to the IL alkali chain length [32]. The degree of crystallinity ($\chi$) also increased [46] depending on the IL chain length, and the mechanical properties changed (plasticizing behaviour in the presence of the IL). Singh et al. [47] conducted a phase, conductivity and dielectric analysis and a morphology study of the $\beta$ phase of PVDF. The objective was to use PVDF as a gel polymer electrolyte for magnesium-ion battery applications. The study reported the presence of a pure $\beta$ phase PVDF membrane and good affinity with a polar organic electrolyte. Yang et al. [48] used a similar approach. They used scanning electron microscope, Fourier transform infrared spectroscopy (FTIR) and X-ray diffraction (XRD) techniques to study the morphology, structure and piezoelectric response of a composite film ($BaTiO_3$/PVDF) for human motion monitoring. As a result, a pressure sensor based on PDA@BTO/PVDF showed a fast response and a good ability to provide an energy supply with an apparent enhancement to output voltages. Despite these successful applications, neither Singh et al. [47] nor Yang et al. [48] considered the inclusion of ILs or used low-toxicity solvents as alternatives to DMF.

Regarding the solvent, DMSO evidences a large spectrum of pharmacological effects, including anti-inflammatory effects, local analgesia and weak bacteriostasis, and it is mainly used as a vehicle for other drugs [49]. It shows a high boiling point (189 °C) and a coefficient of solubility of 16.4 $MPa^{1/2}$ ($\delta_p$), which is close enough to the solute parameter ($\delta_p$ = 12.5 $MPa^{1/2}$). A relative similarity between the Hansen solubility coefficients of a solute and solvent usually indicates ease with dissolving the first into the second. However, other solvents, such as DMF, present a $\delta_p$ closer to that of PVDF, which can represent an advantage in terms of dissolution time. Still, DMSO is a good candidate to dissolve PVDF according to the literature [50,51]. Furthermore, as a solvent, DMSO is used in situations where low toxicity is required and desirable, since conventional solvents are

assumed to be severely toxic and harmful by the European Chemicals Agency [52]. The combination PVDF + DMSO has been used for different purposes: for instance, Wang et al. [53] evaluated the use of DMSO in the process of making lithium ion batteries, while Venault et al. [54] used them to make antifouling membranes. Both authors pointed out the great potential of DMSO as the ideal candidate to replace hazardous solvents.

As a result, with the present study, our research team propose a new experimental protocol to produce a PVDF-based actuator using a non-toxic solvent, which is crucial for biomedical applications. Moreover, the research team expect to better understand the mechanical properties, chemical phase content, as well as the morphology through optical analysis of a composite material based on PVDF by the addition of an IL filler. In the future, we hope the material developed and studied will be integrated into upcoming exosuit solutions, which could be used for aided mobility or rehabilitation scenarios (for instance, for muscle touch feedback for re-educational movement purposes).

## 2. Materials and Methods

### 2.1. Materials

The samples were produced using three commercially available chemical compounds. The PVDF powder (Solef 6020, Brussels, Belgium) was offered by Solvay Lda (Brussels, Belgium). The IL (1-Methyl-3-propylimidazolium bis (trifluoromethylsulfonyl) imide, [PMIM][TFSI], ≥99%), which is responsible for increasing the content of the EA phase in PVDF, was purchased from IoLiTec-Ionic GmbH (Heilbronn, Germany). The polar non-toxic solvent, dimethylsulfoxide (DMSO ANH, 99.8%), was purchased from Fisher Scientific, Lda (Waltham, MA, USA).

### 2.2. Sample Preparation

The samples were produced using an experimental protocol based on Correia et al. [32] and adapted according to the objectives of this work, i.e., non-toxic for biomedical applications. In the first step, the IL was mixed with DMSO, considering a ratio of 40% *w/w* IL/PVDF. In the second step, the PVDF was added to the previous solution at a ratio of 12/88% *w/w* PVDF/DMSO. During the dissolution of the PVDF powder, the beaker was sealed and the mixture was stirred using a magnetic stirrer and heated from room temperature to 50 °C in a thermal bath [55]. After achieving a transparent and homogeneous solution, the resultant mixture was cast to a glass substrate, and the wet film thickness was set to 0.6 mm using a doctor blade technique (Proceq ZUA 2000, Schwerzenbach, Switzerland). On the last step of sample preparation, the wet film was taken into an oven for total solvent evaporation for approximately 30 min. The temperature was set at 85 °C in order to avoid the formation of pores [55]. Figure 1 is a schematic representation of the film production process, as previously described.

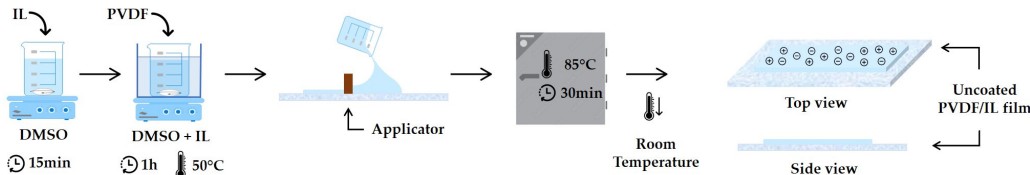

(**a**) Step-by-step production of PVDF-based samples.

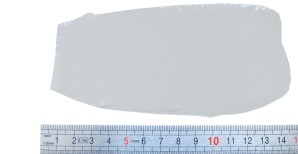

(**b**) Homogeneous and transparent PVDF samples.

**Figure 1.** Schematic illustration of the production (**a**) and the final film (**b**) of PVDF-based samples.

Finally, three specimens with dimensions of $30 \times 10$ mm$^2$ (length $\times$ width) were cut from the sample and weighed. With these measures, the density ($\rho$) of each sample was obtained, and the mean value and standard deviation (STD) were calculated.

### 2.3. Optical Analysis

To attest the porosity level and assess if powder dissolution was completed, the samples were observed using the bright-field microscopy technique. The produced samples were cut into a square shape with dimensions of $12 \times 12$ mm$^2$ and were inspected over a grid composed of smaller squares in a Zeiss apparatus (inverted fluorescence microscope, Axiovert 200M, Oberkochen, Germany) under a 5$\times$ zoom magnification.

### 2.4. Mechanical Characterization

The mechanical tests were conducted following the standard test method for tensile properties of thin plastic sheeting, such as ASTM D882-12 [56]. The samples were cut to dimensions of $75 \times 10$ mm$^2$ (length $\times$ width) and had a thickness of $0.048 \pm 0.004$ mm (mean $\pm$ STD), which is the value expected after solvent evaporation according to Krebs et al. [57]. For the mechanical test, the samples were mounted with a span of 50 mm between grips. The prototype machine used is able to perform uni- and bi-axial tests and is equipped with 120 N actuators and 50 N load cells (Figure 2). The samples were tensile tested leading to rupture at a constant velocity of 5 mm/min (supplementary video). This mechanical approach was used to obtain some of the mechanical properties in the elastic domain, such as the Young's modulus (E), the yield stress ($\sigma_{Yield}$) and the corresponding yield strain ($\varepsilon_{Yield}$).

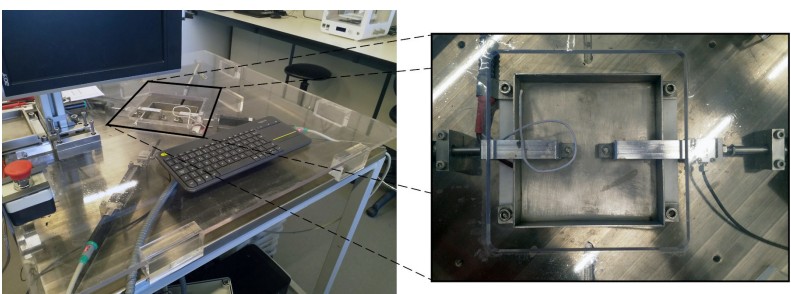

**Figure 2.** Mechanical apparatus used to mechanically tension the samples. At left, we present a general overview of the machine used, while at right, we present a detailed view of the mechanical actuators.

### 2.5. Chemical Characterization

According to Martins et al. [34], the phases of PVDF can be completely characterized by performing different chemical analysis together. FTIR, XRD and differential scanning calorimetry (DSC) were performed to distinguish and identify the prevalent electroactive phase, its content and the degree of crystallinity.

FTIR–attenuated total reflectance (ATR) measurements were conducted using a PerkinElmer FT-IR spectrometer frontier apparatus (Waltham, MA, USA). The analysis was made by considering 64 scans with a resolution of 4 cm$^{-1}$ from 4000 to 600 cm$^{-1}$. From FTIR-ATR, it is possible to distinguish the non-EA phase (e.g., $\alpha$ phase) from the EA phases (e.g., $\beta$ and $\gamma$ phases) present on the material, and from Equation (1) [40], it is possible to calculate the total content of the EA phases (F(EA)).

$$F(EA) = \frac{A_{840,EA}}{\left(\frac{K_{840}}{K_{766}}\right) \times A_{766,\alpha} + A_{840,EA}} \tag{1}$$

where $A_{766}$ and $A_{840}$ are the absorbances at 766 cm$^{-1}$ and 840 cm$^{-1}$, respectively, of the $\alpha$ phase and EA phases, and $K_{766}$ and $K_{840}$ are the corresponding absorption coefficients ($6.1 \times 10^4$ and $7.7 \times 10^4$ cm$^2$/mol, respectively).

Besides FTIR-ATR, XRD (Bruker D8 Advance DaVinci, Billerica, MA, USA) was also used to identify the phases. This technique, in contrast to FTIR, allows clear distinction of the EA phases, such as the $\beta$ phases from the $\gamma$ phases. The method uses a conventional Bragg–Brentano diffractometer, and we adopted the following parameters: wavelength of the incident X-ray beam ($\lambda$) of 1.5405 °A, angle range of $5^\circ \leq 2\theta \leq 45^\circ$, step size of $0.02^\circ$, and 1 s per step.

The combination of both methods (FTIR-ATR and XRD) enables a robust identification of the electroactive phases presented in the material.

DSC measurement was conducted using a Hitachi DSC7020 calorimeter (Hitachi, Ibaraki, Japan) at 10 °C/min from 50 °C to 200 °C. This complementary technique allows us to obtain the degree of cristalinity, $\chi$, of the sample through Equation (2) [32], as well as the melting ($T_m$) and the onset ($T_{onset}$) temperatures from the resultant curve.

$$\chi = \frac{\Delta H}{x\Delta H_\alpha + y\Delta H_{EA}} \tag{2}$$

where $\Delta H$ is the melting enthalpy of the material, $\Delta H_\alpha$ and $\Delta H_{EA}$ are the melting enthalpies of the $\alpha$ (93.07 J/g) and EA phases (103.4 J/g), respectively, and $x$ and $y$ are the $\alpha$ and EA phase proportions, respectively, obtained from FTIR-ATR.

## 3. Results and Discussion

To evaluate the uniformity of the final sample, the density of three different specimens was calculated. The mean value of $\rho$ was 1425.926 $\pm$ 32.075 kg/m$^3$. The low standard deviation ($\approx$2.25%) indicates that all the specimens had similar densities, which means the sample was uniform.

### 3.1. Optical Analysis

Each square sample was segmented into 42 small areas. Figure 3 shows the overall microscopy of an analysed specimen. No porosity was observed in the analysed samples, which indicated a correct solvent evaporation temperature [39]. Moreover, the absence of air bubbles or solvent debris shows that the initially defined evaporation time was adequate for the process. However, the small dots present in the microscopic images might suggest incomplete dissolution of PVDF, indicating that more time would be needed to completely dissolve the PVDF powder and achieve a homogeneous solution at the microscopic level.

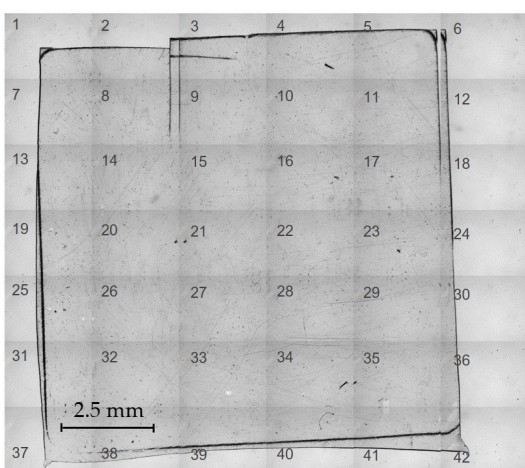

**Figure 3.** Sample viewed under microscope with 5$\times$ magnification.

Gaihre et al. [58] prepared PVDF samples by dissolving 10% PVDF in dimethylformamide (DMF) with different additives and studied the porosity of polypyrrole (PPy)-PVDF micro-actuators. Even though they used a spin coating procedure on a glass side and dried the sample at 50 °C, they obtained samples with no porosity. Poudel et al. [59]

performed digital bright-field microscopy analysis to evaluate the agglomeration of boron nitride nanotubes (BNNTs) in solvent-cast PVDF-trifluoroethylene (TrFE) films, using DMF as solvent. They chose total BNNT dispersion in the samples for lower content of nanotubes. However, the presence of porosity was reported for unpolled and annealed (at 25 °C) samples of PVDF-TrFE and PVDF-TrFE-BNNT 1 wt%. Other authors, such as Chen et al. [60] and Nunes-Pereira et al. [61], suggested that the samples' porosity could be controlled by the solvent choice and by evaporation temperature. However, in both studies, DMF was used to dissolve PVDF.

### 3.2. Mechanical Characterization

The mechanical properties—yield stress, strain and stiffness—were obtained through uniaxial tensile tests. Figure 4 shows the linear elastic behaviour of PVDF samples of 12/88% *w/w* (PVDF/DMSO) from 20 valid tests (gray lines), the mean curve (black full line) and standard error of the mean (SEM) (gray area). Through the mean curve, a $\sigma_{Yield}$ of 9 MPa and a corresponding $\varepsilon_{Yield}$ of 0.06 were estimated. Moreover, a Young's modulus ($E$) of 144 MPa could be obtained from the slope of the linear elastic region. To evaluate the influence of the concentration of PVDF on the mechanical properties, another percentage of PVDF was analysed. Maintaining the proportion of IL/PVDF but changing the ratio of PVDF/DMSO to 15/85% *w/w*, the result of $\sigma_{Yield}$ was 9.8 MPa, and the corresponding $\varepsilon_{Yield}$ was 0.17: both results being higher for this ratio. The value of $E$ was smaller at 118 MPa. This indicates that by increasing the percentage of PVDF, the yield stress and yield strain increase, but the Young's modulus decreases.

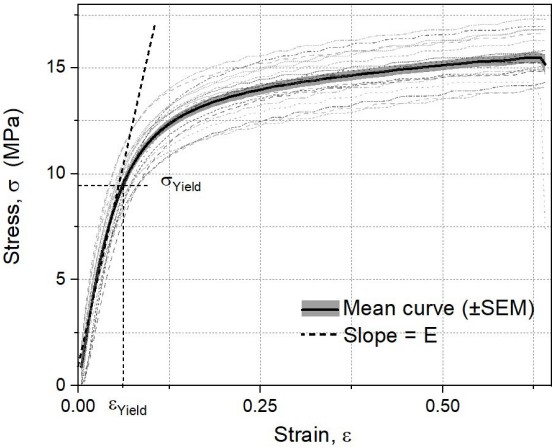

**Figure 4.** Mechanical tensile results of 12/88% *w/w* samples. Grey lines—20 individual specimens; black full line—mean curve; gray area—SEM.

The same relation between the increase of stiffness with the decrease in the percentage of PVDF has also been observed by other authors and has been reported in the literature [62]. Correia et al. [32,46], using DMF as solvent, obtained an $E$ ranging from 75 MPa to 184 MPa and a $\sigma_{Yield}$ varying from 2.53 MPa to 6.4 MPa, with the range of values dependent on the IL used. Comparing the results from Correia et al. [32] with those obtained in this study for 12/88% *w/w* PVDF/DMSO, it is worth pointing out that $E$ is similar and $\sigma_{Yield}$ is on the same order of magnitude. However, the last parameter is higher in the present study, which might be due to different solvents and conditions used in the experimental protocols.

Bao et al. [63] studied a PVDF/IL piezo-active composite film to be used for highly sensitive pressure sensors. The samples were prepared with 1-Ethyl-3-methyl- imidazolium Chlorid ([EMIM]Cl), were dissolved in DMF and were tensile tested, and the stress–strain relation was determined. They reported that the Young's modulus decreased with the IL content. However, Kong et al. [64] studied the mechanical properties of a composite material based on PVDF and reinforced with conductive black carbon and silicon dioxide, since damage severely limits the applications of polymer membranes. The samples were

prepared using DMF as solvent and were tested referring to the ASTM D 882-2012 standard [56]. They conclude that the tensile strength of PVDF increases with the content of the additive.

The results of this study show that the mechanical properties are influenced by the PVDF/DMSO ratio and decrease with the addition of fillers, as reported by Vázquez-Fernández et al. [65]. Nevertheless, the use of a different solvent, when compared to the literature, seems not to have a significant impact on the mechanical properties. Despite the property variations in comparison with PVDF, the piezoelectric material exhibits mechanical properties suitable for real-world applications. Its resistance to plastic deformation at low stress levels can be a potential advantage for applications in rehabilitation scenarios.

### 3.3. Chemical Characterization

FTIR-ATR analysis allowed us to obtain the vibrational absorption band characteristics of each phase ($\alpha$ and EA phases) and the transmittance values needed to calculate the phase content in the sample. Figure 5 shows a peak at 766 cm$^{-1}$, which is representative of $\alpha$ phase, while the peak 840 cm$^{-1}$ validates the presence of EA phases. These results are in agreement with previous studies [32,34] on this type of material. From Equation (1), the content of each phase is calculated. The samples of 12/88% PVDF/DMSO with IL contained approximately 80% EA phase, which is believed to be enough to perform a good piezoelectric response when externally stimulated by electrical inputs. Considering other ratios of PVDF/DMSO (using IL), such as 15/85% *w/w*, the EA content increased to approximately 86%. Without IL, those values decreased to 69% and 70%, respectively.

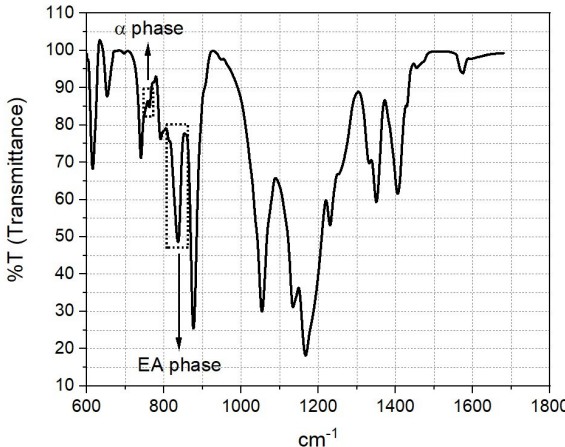

**Figure 5.** FTIR−ATR analysis of the composite material. The peaks corresponding to $\alpha$ and EA phases are represented in the graph and are marked with the dotted rectangles.

The samples with a PVDF/DMSO ratio of 12/88% *w/w* were also analysed with XRD and DSC techniques. XRD analysis allows us to identify and distinguish the $\beta$ phase from the other phases: mainly from the $\gamma$ EA phase. Despite the peak around $2\theta = 20°$ being characteristic of both EA phases of PVDF ($\beta$ and $\gamma$), the absence of clear peaks before that value is an indicator of the main presence of $\beta$ phase instead of $\gamma$ phase (Figure 6) [34,66]. This result is expected for enhanced piezoelectric characteristics of the samples, since the beta phase being the prevalent EA phase is an advantage for piezoelectric materials.

DSC analysis (Figure 7) quantifies the crystallinity of the sample and investigates the response of the polymer to heat. From Equation (2), a degree of crystallinity of approximately 30% was achieved, meaning that 30% of the polymer's mass formed crystalline regions, which directly interfere with some properties, such as the stiffness and piezoelectric behaviour [67]. Higher crystallinity means higher alignment of the polymer chains, which increases the stiffness of the material. The remaining polymer percentage, i.e., 70%, represents the amorphous regions. Moreover, it was possible to determine the melting ($T_m$)

and the onset (T$_{onset}$) temperatures as 159.1 ℃ and 144.68 ℃, respectively. At the end of this analysis, no mass loss was observed.

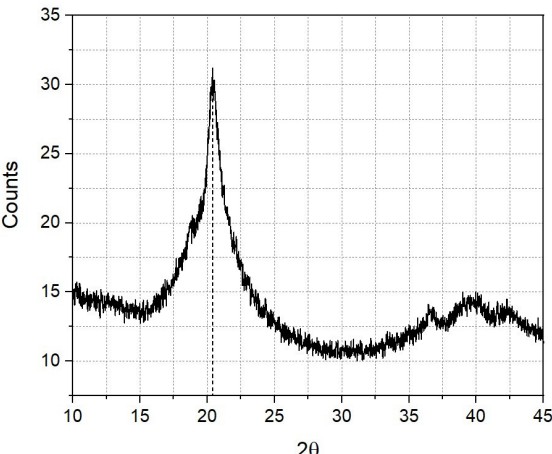

**Figure 6.** XRD analysis of the composite material. The peak at $2\theta \approx 20º$ is characteristic of EA phases.

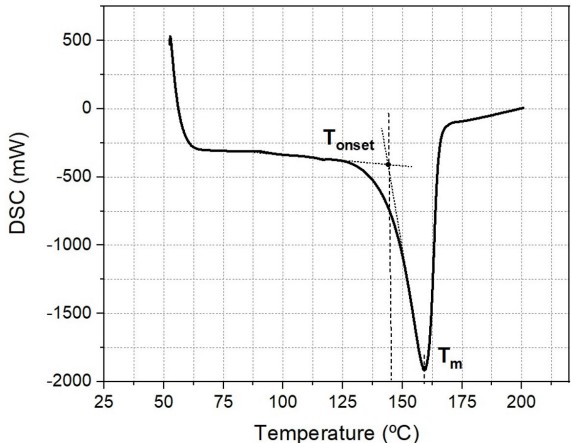

**Figure 7.** DSC analysis of the composite material. Representation of T$_{onset}$ and T$_m$.

Other authors such as Correia et al. [32] and Liu et al. [68] also characterized PVDF films prepared with DMF solvent in terms of chemical properties regarding EA phases. Correia et al. [32] reported a $\beta$ content from 59% to 89% using different types of IL and maintaining the IL/PVDF ratio at 40%. Compared with the results from this study, the values obtained match the range obtained by Correia et al. [32] even though different solvents were used. Liu et al. [68] obtained a $\beta$ content higher than 93% by using the IL ([C2min][BF4]), which is higher than the results obtained in this study. Nevertheless, they also concluded that the polymer phases and their content were influenced by PVDF/solvent ratios and by the presence of IL. Considering the solvent and the presence of IL, a higher concentration of PVDF increased the percentage of electroactive phases.

In addition to the $\beta$ content, Correia et al. [32] also measured the crystallinity and the melting and onset temperatures. They obtained a $\chi$ result varying between 28% and 56% depending on the type of IL used. Regarding the melting temperature, the result obtained via DSC was between 154 ℃ and 173 ℃, while the onset temperature was between 371 ℃ and 407 ℃ as measured by thermal gravimetric analyser (TGA). Begum et al. [69] studied the crystallinity and the thermal stability of PVDF nanocomposites dissolved in DMF and reinforced with epoxy functionalized multi-walled carbon nanotubes (MWCNTs). They found a T$_m$ of 161 ℃ for PVDF. Moreover, although the addition of MWCNTs changed the thermal properties, the authors did not found significant variations in the temperature with an increase in the additives.

Despite the differences between experimental protocols, the results we present in this study compare favourably and are in good agreement with the results found in the literature, regardless of the solvents used.

## 4. Conclusions and Future Works

There are several experimental protocols to produce PVDF-based thin films described in the literature. However, the majority of them use toxic solvents such as DMF or methyl-ethyl-ketone (MEK), which are potentially harmful for users. Therefore, in order to produce an actuator free of toxic solvents (crucial for biomedical applications), in the present study, DMSO was chosen to create a PVDF-based film. Even though several authors have pointed to the possibility of using this solvent, only a few studies can be found in the literature. Hence, it is important to explore the properties of PVDF-based actuators produced by different protocols: especially those using non-toxic solvents. Having non-toxicity and strong piezoelectric behaviour as principal objectives in this study, our research team found an optimal ratio of 12/88% $w/w$ PVDF/DMSO to obtain a more fluid mixture that facilitates the dissolution of the PVDF powder. Moreover, the best temperature for PVDF dissolution was also successfully achieved. A very low temperature would increase the dissolution time, while a high temperature could degrade the material. Therefore, a temperature of 50 ºC was the ideal value to develop a suitable experimental protocol.

Characterization of the material revealed that the inclusion of IL changes all the properties analysed, such as the Young's modulus ($E$), the yield stress ($\sigma_{Yield}$) and the $\beta$ phase content. Nevertheless, considering similar ratio conditions, the use of a different solvent does not evidence deep changes to the material's properties, which points to the possible use of DMSO as a replacement solvent for hazardous alternatives. In the future, we aim to study the electro–mechanical properties and the response of the PVDF-based smart material to investigate even further its use in biomedical applications. Given the promising results obtained for the properties of this polymeric material, our research team intends to develop a biomedical device using a piezoelectric PVDF thin film as the main element.

With this study, our research team hopes to have contributed significantly to the knowledge about piezoelectric PVDF-based materials and the potential use of these smart soft materials in future biomedical applications.

**Supplementary Materials:** The following supporting information can be downloaded at: https://www.mdpi.com/article/10.3390/app14083356/s1, Supplementary video: Tensile test (sped up 15 times).

**Author Contributions:** Conceptualization, A.D.A. and P.M.; investigation, A.D.A.; methodology, A.D.A.; formal analysis, A.D.A. and A.M.T.; writing—original draft, A.D.A.; writing—review and editing, A.D.A., A.M.T. and P.M.; supervision, P.M. All authors have read and agreed to the published version of the manuscript.

**Funding:** António Diogo André (A.D.A) gratefully acknowledges funding from Fundação para a Ciência e a Tecnologia (FCT), Portugal, under grant SFRH/BD/147807/2019. Ana Margarida Teixeira (A.M.T.) acknowledges grant 2020.08718.BD; Pedro Martins (P.M.) gratefully acknowledges funding from FCT through INEGI under LAETA, project UIDB/50022/2020.

**Data Availability Statement:** The raw data supporting the conclusions of this article will be made available by the authors on request.

**Conflicts of Interest:** The authors state that they have no financial, professional or other personal involvement in any product, service and/or company that would possibly affect their stance.

**Abbreviations**

The following abbreviations are used in this manuscript:

| | |
|---|---|
| DMAC | Dimethylacetamide |
| DMF | Dimethylformamide |
| DMSO | Dymethilsulfoxide |
| EA | Electroactive |
| DSC | Differential scanning calorimetry |
| FEUP | Faculty of Engineering of the University of Porto |
| FTIR | Fourier-transform infrared spectroscopy |
| i3A | Aragón Institute for Engineering Research |
| IL | Ionic liquid |
| INEGI | Institute of Science and Innovation in Mechanical and Industrial Engineering |
| LAETA | Associated Laboratory of Energy, Transport and Aeronautics |
| MEK | Methyl ethyl ketone |
| MWCNTs | Multi-walled carbon nanotubes |
| PVDF | Polyvinylidene fluoride |
| SEM | Standard error of mean |
| STD | Standard deviation |
| UBS | Biomechanic and Health Unit |
| XRD | X-ray diffraction |

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
