# Peer review of "Influence of DMSO Non-Toxic Solvent on the Mechanical and Chemical Properties of a PVDF Thin Film"

_applsci, doi:10.3390/app14083356_

Round 1

Reviewer 1 Report

Comments and Suggestions for Authors

The method presented to create a PVDF-based film using a less toxic solvent (DMSO)and a ionic liquid is promising for future applications. However, DMSO has some drawbacks (it evaporates slowly at normal atmospheric pressure) which should be taken into consideration for practical applications.

Author Response

Please, see the file attached

Reviewer 2 Report

Comments and Suggestions for Authors

This manuscript discusses about using a non toxic solvent for preparing the piezoelectric thin films of PVDF. Therefore, it matches with the scope of Applied Sciences.

However, some questions and concerns still need to be addressed:

1.) Page 1, the title: I recommend the authors directly point out 1.) what the non toxic solvent is used in this study; 2.) what the piezoelectric thin film is investigated here in this study.

2.) Page 1-2: the introduction: there are a lot of discussions on the functionalities of PVDF, however, all these seem not directly relevant to this manuscript. If the solvent is the subject of this study, I recommend the authors provide more information on the solvent and its development for PVDF.

3.) Page 5, line 168: the authors claim that no porosity is observed, could the authors comment on could this be due to the resolution of optical microscopy is not high enough ? Could the authors comment on what size of the porosity is a concern ?

4.) In general, the title emphasizes the importance of the solvent on PVDF mechanical and chemical properties. Therefore, it is very important to compared the same properties of PVDF when using other solvents to this study. However, there are some comparison in this manuscript by mentioning the authors of other works but there are no discussions 1.) what solvent other works used; 2.) what are the commons and differences when using different solvents and their impact. Could the authors add these info ?

Based on the current status of this manuscript, I recommend a minor revision.

Comments on the Quality of English Language

English quality is fine in general, only minor editing is needed.

Author Response

Please, see the file attached

Reviewer 3 Report

Comments and Suggestions for Authors

Comments:

In the article "Influence of a Nontoxic Solvent on the Mechanical and Chemical Properties of a Piezoelectric Thin Film," the authors developed a protocol for making piezoelectric PVDF-based materials using the nontoxic solvent DMSO and a specific ionic liquid. The manuscript is written and well-explained. However, there are a few points in the paper that require clarification before publication.

On page 1, line 35, the authors mention that "PVDF has a large dielectric constant, high polarity," etc. It would be beneficial if the authors could provide the actual value of the dielectric constant (xx) at a specific temperature and frequency, as well as other relevant properties. This would enhance the understanding of PVDF properties.

On page 2, line 50, the author mentions that "DMSO can also play a role in β-phase formation." It would be helpful if the author briefly explained the specific role that DMSO plays in this process.

On page 2, lines 75-77, the author should explain the claims reported in reference 47, which proposed DMSO as an alternative solvent previously. Additionally, they should clarify the improvements proposed in this work compared to previous research.

On page 2, line 78, instead of "the authors propose a new experimental," it would be clearer to use "we propose" to indicate the researchers' involvement.

Throughout the paper, there are instances where the authors use phrases like "the authors expect" or "the authors propose." It may be more appropriate to use "we" or "our research team" to clarify that the paper reflects the work of the researchers.

On page 3, line 92, the author writes "non-toxic," but in the title, it is "nontoxic." The author should maintain consistency throughout the manuscript.

In Figure 2 on page 4, the author should explain in the caption of what is shown in the figure.

The author mentions Section 3 as "Results and Discussion," but the subsections are labelled as "3.1 Optical Results," "3.2 Mechanical Testing Results," etc. It would be clearer to either add a separate subsection for discussions or modify the subsection names accordingly.

Based on these points, I recommend minor revisions before publication in Appl. Sci. in the current form of the manuscript.

Author Response

Please, see the file attached
